# Magnetic Properties Regulation of FeGa and FeGaNi Films with Oblique Magnetron Sputtering

Chun Wang [1], Sanmin Ke [2] and Zhen Wang [2,*]

1    College of Science, Xi'an Shiyou University, Xi'an 710065, China
2    Department of Applied Physics, Chang'an University, Xi'an 710064, China
*    Correspondence: wangzhen@chd.edu.cn

**Abstract:** Magnetic FeGa and FeGaNi films with an in-plane anisotropy were deposited by employing oblique magnetron sputtering. With the increase in oblique angle, the crystallite size of FeGa decreases, which indicates that oblique sputtering can refine the crystallite size. The remanence ratio of FeGa films increases from 0.5 to 0.92 for an easy axis, and the coercivity increases with the decrease in the crystallite size. The calculated static anisotropic field shows that the in-plane magnetic anisotropy can be induced by oblique sputtering and the strength increases with the oblique sputtering angle. After doping Ni by co-sputtering, FeGaNi films exhibit a stable remanence ratio at 0.8, low coercivity and good anisotropy. With the low sputtering power of the Ni target, there is a competitive relationship between the effect of crystallite size and Ni doping which causes the coercivity of FeGaNi films to first increase and then decrease with the increase in the oblique angle. The FeGaNi film also shows high anisotropy in a small oblique angle. The variation of coercivity and anisotropy of FeGaNi films can be explained by the crystalline size effect and increase in Ni content. For the increasing intensity of collisions between FeGa and Ni atoms in the co-sputtering, the in-plane magnetic anisotropy increases first and then decreases. As a result, the magnetic properties of FeGa films were examined to tailor their magnetic softness and magnetic anisotropy by controlling the oblique sputtering angle and Ni doping.

**Keywords:** FeGa film; FeGaNi film; oblique magnetron sputtering; magnetic property; magnetic anisotropy

## 1. Introduction

Iron–gallium alloys ($Fe_{100-x}Ga_x$, x from 13 to 29 at %), as a fascinating magnetostrictive material, have been confirmed to possess moderate magnetostriction (about 400 ppm for Ga content at 19%) in a low magnetic field as well as good mechanical properties [1–4]. FeGa, with its bcc structure and crystallographic orientation to (110), is usually preferred for obtaining good magnetostrictive properties [5]. This has attracted widespread studies focusing on FeGa materials for strain-mediated magnetoelectric devices [6–10]. Meanwhile, compared with traditional magnetostrictive materials, such as TbDyFe, FeGa shows soft magnetic properties. FeGa exhibits a fantastic potential application in fabricating more efficient, faster, and smaller high-frequency devices in information and communication technologies, for instance, recording heads, wireless inductor, and microwave noise filters [11–14]. However, the $H_c$ of the FeGa film is relatively high and the $M_r/M_s$ could be less than 0.7 [15–17]. Such properties could suppress the microwave magnetization precession of FeGa films, which go against high-frequency applications.

Therefore, researchers have used various methods to optimize the soft magnetic properties of the FeGa film. To achieve a good high frequency permeability, the material should possess low coercivity ($H_c$) and a high remanence ratio ($M_r/M_s$), which is used in high-frequency applications [11,12]. The FeGa/Magnetic heterostructures such as FeGa/FeNi [18], FeGa/Mn [19], $Fe_{75.5}Cu_1Nb_3Si_{13.5}B_7$/FeGa [20], FeGa/Cu [21], FeGa/Ta [22], FeGa/diamond [23], and FeGa/ZnSe [11] were used to improve the soft

magnetic properties through the interfacial interaction, the number of layers, and their thicknesses. For the heterostructure of hard and soft ferromagnets in films, a strong exchange spring magnetism assumes the responsibility of explaining the enhanced static and dynamic magnetic properties. The coupling between the hard ferromagnetic layer and the soft ferromagnetic layer reduces the energy required to flip the magnetization of the composite [24–26]. The remanent magnetization and the exchange bias of heterostructures could be remarkably changed by means of modifying the strength and the orientation of the uniaxial magnetic anisotropy which makes them extremely promising candidates for high-frequency, strain-coupled multiferroic systems [27,28]. Otherwise, the element doping FeGa films also proved to be an effective approach to optimize the soft magnetic properties of FeGa films by reducing the crystallite size [15,29]. The magnetic anisotropy of the magnetostrictive FeGa film, which is one of the vitally important properties, can be obtained by mechanical stress because of the magnetoelastic coupling in the film, and then alters the magnetostriction of the FeGa film [16,30].

The in-plane uniaxial anisotropy of magnetic films can be effectively induced by the growth of the oblique sputtering, which has been proven to be an effective method [31]. In this work, oblique sputtering was employed to fabricate FeGa and FeGaNi films at room temperature. By changing the oblique sputtering angle, the in-plane magnetic anisotropy of the films can be controlled. The microstructure and soft magnetism of the films were investigated. The addition of an Ni element was used to enhance the soft magnetic phase. There are some reports about the heterostructure of FeGa/FeNi [18,32], and the FeGa/FeNi heterostructure was used to improve the soft magnetic properties through the interfacial interaction. However, we can hardly find some reports about Ni-doped FeGa films. As a result, the FeGa films showed significant in-plane magnetic anisotropy and FeGaNi films also enhance the soft magnetic properties of FeGa films.

## 2. Experimental

FeGa and FeGaNi films were grown on $25 \times 25$ mm$^2$ silicon (111) substrates by using DC magnetron sputtering (Kurt J. Lesker) with different oblique sputtering angles. The composition of the target was fixed at Fe$_{80}$Ga$_{20}$ (at%), and all the targets had a diameter of 2.0 inches. The base pressure for deposition is less than $1 \times 10^{-7}$ Torr and the films were deposited in a 5 mTorr Ar atmosphere. The sputtering power was set to 48 W for FeGa target and 10 W for the Ni target. The schematic diagram of the sputtering process is shown in Figure 1. The center distance of the FeGa and Ni targets is approximately 180 cm. The Si substrate is placed in a line which is approximately158 cm above the center connection of the FeGa and Ni targets. α and β are the oblique angles of the FeGa and Ni targets with the substrate, respectively. In order to obtain a different oblique sputtering angle, the substrate was placed in such a position in order of 30, 60, 90, 120, and 160 cm along the x axis, and the corresponding angles α and β are shown in Table 1. It can be found that the variation of angles α and β show opposite tendencies over position x. Thus, herein, we only need to consider the case of the variation of angle α. Then, we fabricated FeGa films with the FeGa target and prepared FeGaNi films with the FeGa and Ni targets. The sputtering rates of the FeGa and Ni targets vary with different x positions through the surface profile meter. By controlling the sputtering switch and sputtering time, a series of FeGa and FeGaNi films with a thickness of approximately 85 nm was obtained.

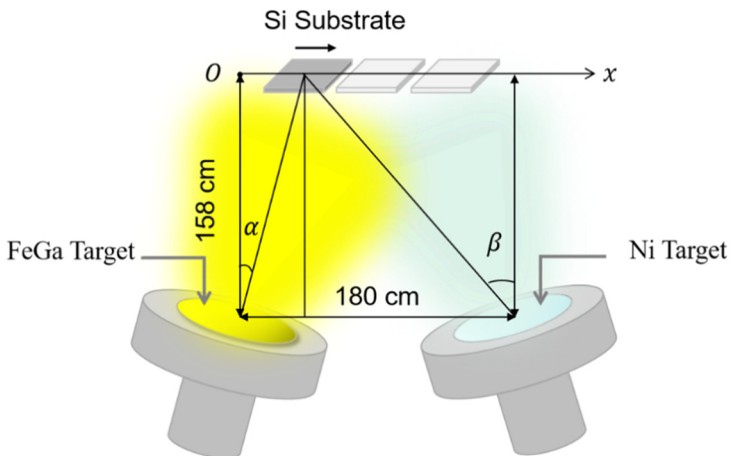

**Figure 1.** The schematic diagram of co-sputtering with different oblique sputtering angles.

**Table 1.** The position and corresponding angles α and β.

| x (cm) | 30 | 60 | 90 | 120 | 150 |
|---|---|---|---|---|---|
| α | 12.5° | 24° | 33.7° | 41.6° | 48° |
| β | 48° | 41.6° | 33.7° | 24° | 12.5° |

The crystal structures were analyzed by X-ray diffraction (XRD, PANalytical X'Pert) with a Cu Ka source ($\lambda$ = 1.5406 Å). The elemental compositions of the films were carried out by an energy-dispersive X-ray spectrometer (EDX, Joel 6610). The thicknesses of the films were performed with a Surface Profile-meter (Dektak 8). The hysteresis loops of FeGa and FeGaNi films were measured by vibrating the sample magnetometer (VSM, Lakeshore 7304) at room temperature, and the magnetic field was applied in parallel to the film plane.

## 3. Experimental Results and Discussion

The structural characterization of the FeGa and FeGaNi films were investigated by using XRD. The oblique sputtering angle α dependent XRD spectra for FeGa and FeGaNi films are shown in Figure 2a,b, respectively. It can be clearly seen that all the films display a strong diffraction peak at 44.6° corresponding to the bcc (110) diffraction peak and two weak peaks at 65° and 82° corresponding to the bcc (200) and (211) diffraction peaks for FeGa or FeGaNi films, illustrating a common polycrystalline structure. The strength of the peaks indicates that the growth direction is a slight preferred orientation along (110) [9,32,33]. The position and shape of the peaks have no clear change. The strength of the peaks decreases with the increase in the α angle. This is because the distance between substrate and targets changes with the angles which is an important reason to change the quality of the film. Meanwhile, in the same oblique sputtering angle, the strength of the peaks of FeGaNi is stronger than that of FeGa. For the Ni film, a strong peak of (111) is at approximately 44.6° and a weak peak (200) is at approximately 52° [34,35]. However, from Figure 2b, there is no peak at approximately 52°. This may be because that the low content of Ni causes the (200) peak not to appear in FeGaNi films. Thus, it is difficult to distinguish FeGa and Ni from the XRD patterns.

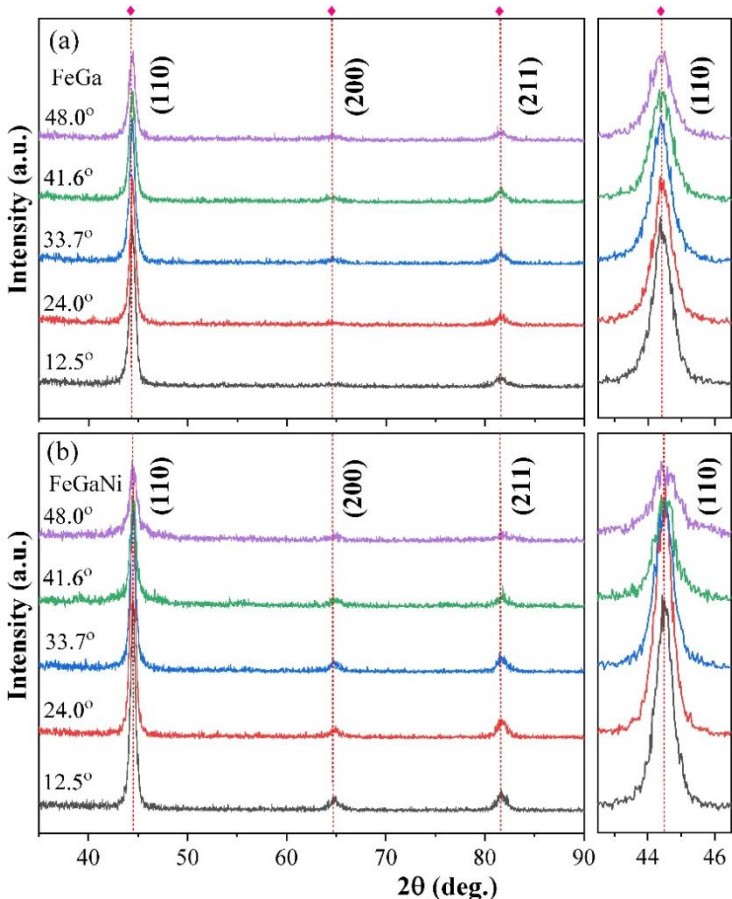

**Figure 2.** XRD spectra of FeGa (**a**) and FeGaNi (**b**) with increase in α.

According to Scherrer's formula and the full width at half-maximum (FWHM) of the (110) peak, for FeGa films, the crystallite size gradually decreases from 25.6 to 22.6 nm with increase in the α angle. This indicates that oblique sputtering can refine the crystallite size. While for FeGaNi, the crystallite size decreases from 28.7 to 17.1 nm with increase in α angle in Figure 3. When the α angle is smaller than 41.6°, the crystallite size of FeGaNi is larger than that of FeGa. In a larger α angle, the Ni content increases and plays the main part, which cause the crystallite size of FeGaNi to be smaller than that of FeGa. The compositions of the FeGaNi films were determined by EDX. As shown in Table 2, the EDX results show that the composition of Ni increases from 1.66% to 5.96% with the increase in α angle from 12.5° to 48.0°. With the low sputtering power of Ni target, the content of Ni in FeGaNi films is small. Meanwhile, from Table 2, the composition of Fe and Ga decreases from 84.75% to 82.75% and 13.59% to 12.3% with the increase in α from 12.5° to 48.0°, respectively. This is because, in the co-sputtering process, the distance between the substrate and two targets is different, which causes a varying sputtering rate and induces different proportions of elements in FeGaNi films with the position of substrate moving from left to right. Thus, the distance between the substrate and FeGa target increases with the increase in α, which causes a decrease in the Fe and Ga content in the films. In contrast, the distance between the substrate and Ni target decreases with the increase in α, which causes an increase in Ni content. However, the atomic ratio of Fe to Ga remains at approximately 6.2 with the increase in α angle in FeGaNi samples which is larger than that of FeGa target (the atomic ratio of Fe to Ga is 4). The result is coming from that the sputtering rate for a different element is very different in the sputtering process. The composition of Ni is much less than FeGa in FeGaNi films that the sputtering power of the Ni target is much less than that of FeGa target in the sputtering process. The magnetic properties of the FeGa film could be influenced by the crystallite size and the content of Ni doping.

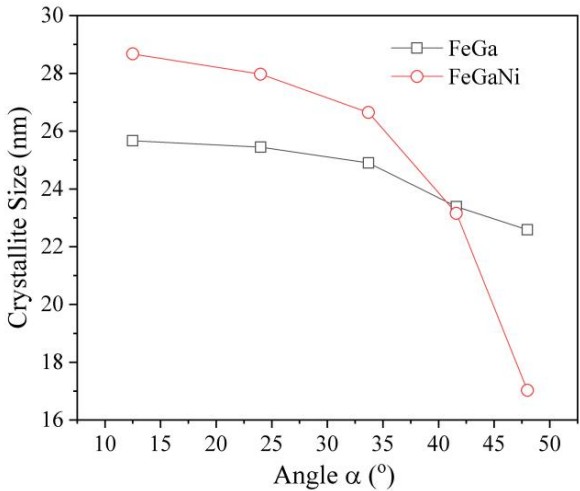

**Figure 3.** Crystallite size of FeGa and FeGaNi films calculated by Scherrer's law with FWHM of (110) peak.

**Table 2.** Chemical composition of FeGaNi film with α.

| A | 12.5° | 24° | 33.7° | 41.6° | 48° |
|---|---|---|---|---|---|
| **Fe (%)** | 84.5 | 84.25 | 83.56 | 83.06 | 82.75 |
| **Ga (%)** | 13.59 | 13.62 | 12.8 | 12.64 | 12.3 |
| **Ni (%)** | 1.66 | 2.31 | 3.64 | 4.31 | 5.95 |
| **Fe:Ga** | 6.24 | 6.19 | 6.53 | 6.57 | 6.65 |

The typical hysteresis loops of FeGa and FeGaNi films corresponding to the increase in α angle are shown in Figure 4. The loops of FeGa films along the easy and hard axis are almost the same when the α angle is less than 24°. However, the magnetic anisotropy becomes obvious with the increase in the α angle. This indicates that in-plane anisotropy can be induced by oblique sputtering for the internal stress in FeGa films [9,31]. Meanwhile, in Figure 4b, FeGaNi films show narrower hysteresis loops and relatively stronger in-plane magnetic anisotropy compared with FeGa, which means that doping Ni can improve the soft magnetic property of FeGa films. However, the in-plane magnetic anisotropy increases when angle α is less than 33.7° and then decreases as angle α increases. This may be explained by the collisions between FeGa and Ni atoms in the sputtering process. From Figure 1, the distance between the substrate and Ni target decreases with the increase in angle α, so the effect of collisions between FeGa and Ni atoms would increase. Meanwhile, with the low sputtering power of Ni target, the distance between the substrate and Ni target is relatively large when the α angle is less 33.7°, and the effect of collisions between the FeGa and Ni atoms is not obvious in the formation of in-plane magnetic anisotropy. However, as the α angle continues to increase, the distance between the substrate and Ni target reduces, the effect of collisions between FeGa and Ni atoms becomes obvious. Then, the in-plane magnetic anisotropy decreases.

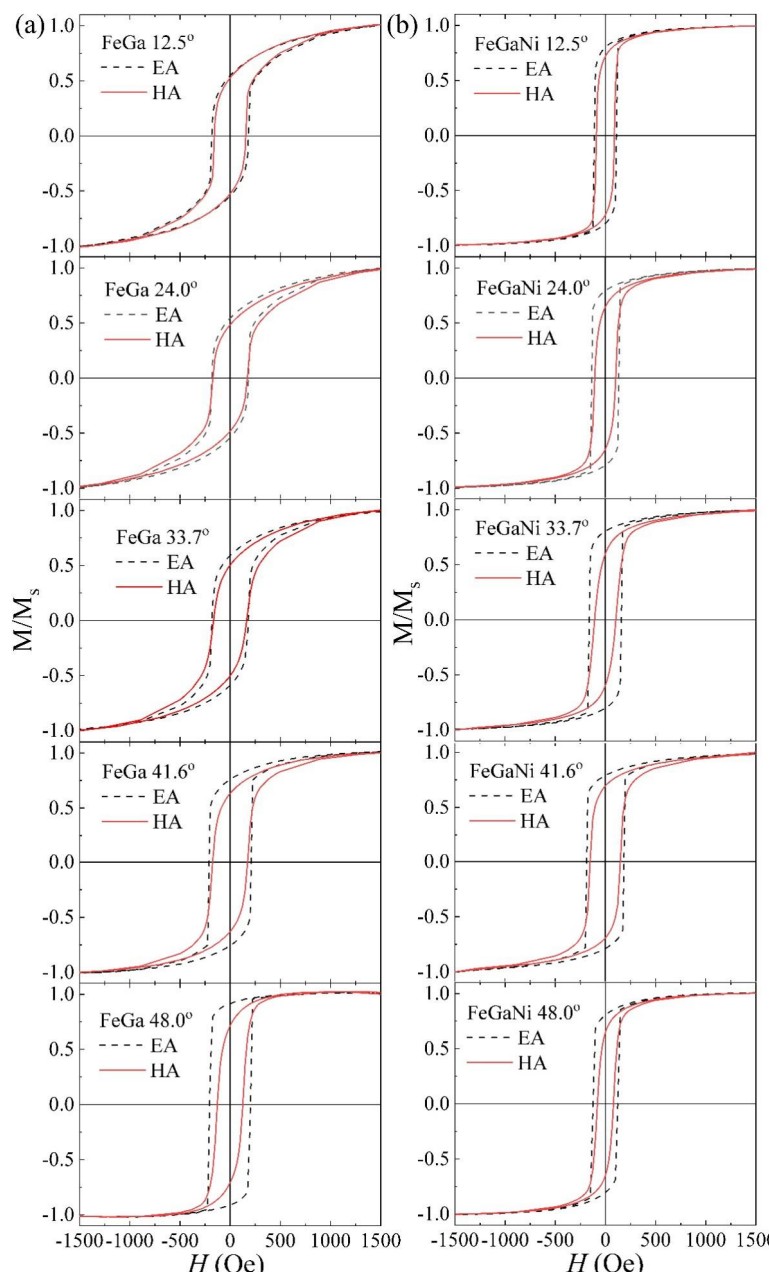

**Figure 4.** Hysteresis loops of FeGa (**a**) and FeGaNi (**b**) with increase in α.

From Figure 5a, the remanence ratio ($M_r/M_s$) of FeGa films gradually increases from 0.5 to 0.92 with the increase in angle α. The enhanced $M_r/M_s$ by oblique sputtering is very obvious [9], while the $M_r/M_s$ of FeGaNi films remains at approximately 0.8 with the increase in angle α. This indicates that the $M_r/M_s$ of FeGa films can be improved by doping Ni when angle α is less than 41.6°. In Figure 5b, the coercivity of the easy axis ($H_c$) of FeGa films decreases from 183 to 180.5 Oe when angle α is less than 33.7° and then increases to 208 Oe when angle α is 41.6° and 204 Oe when angle α is 48°. The change I the coercivity of FeGa films could be understood by the change of crystallite size. When the crystallite size is larger than the exchange length, the coercivity would increase with the decrease in crystallite size for soft magnetic materials [36,37]. The exchange length of FeGa is 5.7 nm [36,37], which is smaller than the crystallite size of FeGa from Figure 3. The crystallite size of FeGa gradually decreases with the increase in angle α from Figure 3. When angle α is less than 33.7°, the crystallite size of FeGa shows small change. The coercivity of FeGa films also expresses very little change. As angle α continues to increase,

the coercivity of FeGa increases and the crystallite size decreases, which is similar to what was expected. Comparing the coercivity of the FeGaNi and FeGa films in Figure 5b, the coercivity of FeGaNi films is smaller than that of FeGa films, which indicates that the doping is an effective means of improving the soft magnetic property. However, the $H_c$ of FeGaNi films first increases from 110 to 180 Oe, when angle $\alpha$ increases from 12.5° to 41.6° and then decreases to 120 Oe when angle $\alpha$ is 48°. From Figure 3, the crystallite size of FeGaNi decreases with the increase in angle $\alpha$. Thus, the coercivity of FeGaNi films should increase with the increase in angle $\alpha$. However, as angle $\alpha$ increases, the content of Ni also increases, as can be seen in Table 2. The soft magnetic phase FeNi may be formed in FeGaNi films. As a result, there is a competitive relationship between the effect of crystallite size and Ni doping on the coercivity of FeGaNi films. So, when angle $\alpha$ is less than 33.7°, the crystallite size effect takes the main responsibility for the increase in $H_c$. With the increase in angle $\alpha$, the effect of Ni doping becomes competitive, which could lead to the decrease in coercivity when angle $\alpha$ is 48°.

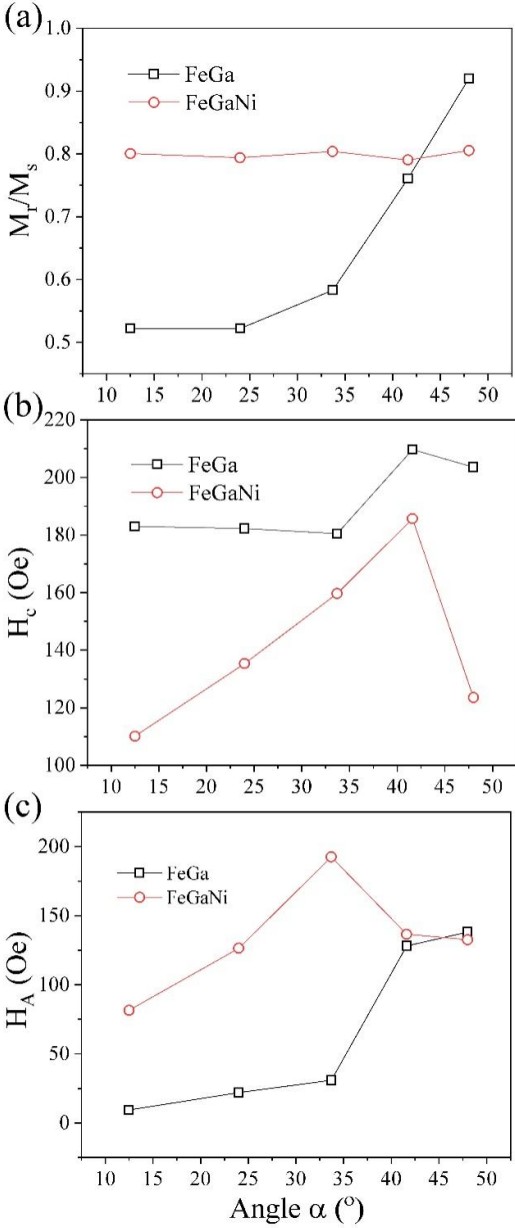

**Figure 5.** Mr/Ms (**a**), Hc (**b**), and HA (**c**) of FeGa and FeGaNi with increase in $\alpha$.

To determine the static anisotropic field $H_A$, we can utilize the area enclosed by the normalized magnetization curve and the horizontal axis and calculate the difference from the area between the easy and hard axis [38,39].

$$H_A = 2 \int_0^{H_{up}} [m_{EA}(H) - m_{HA}(H)] dH \tag{1}$$

where the upper integration boundary $H_{up}$ is any value greater than the saturation field, and $m_{EA}$ and $m_{HA}$ are the easy axis and hard axis loops of the reduced magnetization ($M/M_s$), respectively. The calculated $H_A$ is shown in Figure 5c. The $H_A$ of FeGa films increases from 9.4 to 31 when angle $\alpha$ is less than 33.7°, then rapidly increases from 31 to 128.1 Oe as the angle $\alpha$ increases from 33.7° to 41.6°, and then increases to 138.2 Oe at 48°. This shows that the obvious anisotropy of FeGa films can be induced by tuning an oblique sputtering angle. However, the $H_A$ of FeGaNi films first increases from 81.1 to 192.6 Oe when angle $\alpha$ increases from 12.5° to 33.7°, and then decreases to 132 Oe when angle $\alpha$ is 48°. The anisotropy of FeGaNi films is larger than that of FeGa when angle $\alpha$ is less than 41.6°. This indicates that strong anisotropy can be induced in FeGa films by doping Ni in this method. The decrease in $H_A$ of FeGaNi films when the $\alpha$ angle is larger than 33.7° may be explained by collisions between FeGa and Ni atoms in the sputtering process. With the low sputtering power of the Ni target, the distance between the substrate and Ni decreases with the increase in angle $\alpha$, the intensity of the collision between FeGa and Ni atoms increases, which could reduce the in-plane magnetic anisotropy.

## 4. Conclusions

FeGa and FeGaNi films were successfully fabricated by oblique magnetron sputtering on Si (111) substrates. By controlling the oblique sputtering angle, the magnetic properties of FeGa and FeGaNi films were examined to tailor their magnetic softness and magnetic anisotropy. With the increase in the oblique sputtering angle, the remanence ratio of FeGa films gradually increases from 0.5 to 0.92, and the coercivity increases with the decrease in the crystallite size. The calculated static anisotropic field shows that the in-plane magnetic anisotropy can be induced by oblique sputtering and the strength increases with the oblique sputtering angle. After doping Ni by co-sputtering, the variation of oblique angle FeGaNi films exhibits a stable remanence ratio of 0.8, low coercivity and good anisotropy. With the low sputtering power of the Ni target, there is a competitive relationship between the effect of crystallite size and Ni doping which causes the coercivity of FeGaNi films to increase at first and then decrease with the increase in the oblique angle. The FeGaNi film also shows high anisotropy in the small oblique angle. The variation in the coercivity and anisotropy of FeGaNi films can be explained by the crystalline size effect and increase in Ni content. Due to the increase in the collision intensity between FeGa and Ni atoms in the co-sputtering, the in-plane magnetic anisotropy first increases and then decreases.

**Author Contributions:** C.W., investigation, writing—original draft, and data curation; S.K., investigation and data curation; Z.W., supervision, conceptualization, methodology, and project administration. All authors have read and agreed to the published version of the manuscript.

**Funding:** This work was supported by Shaanxi Provincial Natural Science Basic Research Program (2013JQ1011), Shaanxi Provincial Key Laboratory of Frontiers in Theoretical Physics Open Fund Project (SXKLTPF-K20190603).

**Institutional Review Board Statement:** Not applicable.

**Informed Consent Statement:** Not applicable.

**Data Availability Statement:** Data available on request from the corresponding author.

**Conflicts of Interest:** The authors declare no conflict of interest.

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
