# Peer review of "Magnetic Properties Regulation of FeGa and FeGaNi Films with Oblique Magnetron Sputtering"

_magnetochemistry, doi:10.3390/magnetochemistry8100111_

Round 1

Reviewer 1 Report

The article "Magnetic properties regulation of FeGa and FeGaNi thin films with oblique magnetic sputtering " presents the structural and magnetic results of FeGa and FeGaNi thin films obtained by oblique magnetic sputtering. Although a great effort was used to obtain these results, I do not consider that the paper can be published in the present form. The English language needs extensive editing. The results are not clearly presented. It is not clear what is the novelty of the study. There are other reported results on FeGa and FeGaNi thin films obtained by oblique magnetic sputtering. What are the new information that the authors bring to this research topic? This should be clearly specified in the introduction section. 

Moreover, here are other aspects which have to be reconsidered: 

- the  Scherrer’s formula does not include the grain size but the crystallite size. 

- since the compositional ratio Fe:Ga is different from the one in the target, does that mean that the deposition method used does not ensure a stoichiometric transfer?

- how did the authors concluded that the films present an in-plane magnetic anisotropy since the magnetic field in the VSM measurements was only applied on the parallel to the surface direction. Are there any out-of-plane hysteresis loops recorded? How were the easy axis and hard axis established? It would be of great interest to present the magnetization values since the thickness of the sample was obtained by profilometry. How was the magnetic exchange length estimated? The specified references do not include values for these specific materials. 

It is important that the authors compare their results with the ones reported by other research groups. In the "Experimental Results and Discussion " section only 4 references are mentioned and these refer to the HA equation and to magnetic exchange lengths. 

The explanation on the coercive field variation are not clearly presented. Please rephrase.

Author Response

Thanks very much for your comments on this paper. Please find the reponse in the attachment.

Reviewer 2 Report

Magnetic properties of FeGa and FeGaNi thin films investigated by Zhen Wang and coauthors demonstrate very nicely. Controlling oblique sputtering angle, the magnetic properties of thin film were examined and well correlated and explained very well. Before going for the publication, author should address some minor comments on this work. Please go through these comments carefully:

Comment #1: The condition for co-sputtering of thin thin film are maintained. I just think, if authors can explain better that, how and why the angles affects the magnetic properties of thin film, I will appreciate.

Comment #2: I see the nice XRD spectra of both film. The (110) diffraction peak change with angles. But I am curious, that changes might be due to other reasons, such as the distance between substrate and gun when you change the angles, which change the quality of the thin film. I will suggest authors to explain this in the manuscript.

Comment #3: From the Figure 1, I expect, when b is larger, the (110) plane should be sharp. But I see other way. Please correct me if I am wrong or explain this in the manuscript.

Comment #4: Anisotropy was observed due to the larger sputtering angles. Some places it is written it is due to smaller oblique angles. It is confusing to read and get the connectivity in the findings. I suggest authors to take care them.

Thank you very much. I really appreciate all the authors for this nice piece of work.

Author Response

(The authors gave the same response as above.)

Reviewer 3 Report

Dear Authors,

In this work, you studied the effect of doping with nickel on the magnetic properties of FeGa films. I read your Manuscript with interest.

In my opinion, to improve the quality of the article, it is desirable to take into account the following comments:

Main comments:

It is important for the reader to understand the novelty and the goal of the work.  It is impossible to understand them from the text of the article. For example, Introduction states that “…researchers have used various methods to optimize the soft magnetic properties of FeGa films…” (line 15). Further, various “heterostructures” are described in detail and only one article is noted “…the element doping to FeGa thin films…” Although other publications are known, for example, He, Y., Jiang, C., Wu, W., Wang, B., Duan, H., Wang, H., … Xu, H. (2016). Giant heterogeneous magnetostriction in Fe–Ga alloys: Effect of trace element doping. Acta Materialia, 109, 177–186 or Meng, C., Wu, Y., & Jiang, C. (2017). Design of high ductility FeGa magnetostrictive alloys: Tb doping and directional solidification. Materials & Design, 130, 183–189.

In this regard, there is a number of questions which are desirable to answer in the text of the Manuscript:

1. Why was the element doping method chosen to optimize the soft magnetic properties of FeGa films?

2. Why was nickel chosen for element doping to FeGa films?

3. What is the novelty of the work?

4. What was the goal of the Authors?

5. How did the magnetic properties of FeGaNi films improved compared to FeGa or FeGaX (X is a doping element)?

Additional comments:

1. The use of the term “thin film” requires an indication of possible size effects (tunneling, phase transitions, etc.). If there are none, then the film cannot be called "thin";

2. What is the meaning of the term “mechanically magnetic” (line 188)?

Author Response

Thanks very much for your comments on this paper. Please find the response in the attachment.

Reviewer 4 Report

Dear Authors, your work must be seriously improved according to next observations.

1. Title: Magnetic or Magnetron sputtering?

2. Do not insert abbreviations in text without explaining them: Abstract: (Mr/Ms means what?) Check the all manuscript for such problems.

3. Keywords: I suggest you to change “oblique sputtering” with “oblique magnetron sputtering”.

4. Check the all manuscript for English translation errors, such as “FeGa exhibit” or “aplications of information and communication technologies” from Introduction chapter. Or the “insect” from Figure 3 caption. And so on.

5. Please insert in the Introduction chapter the novelty of your work, what brings new comparatively to the state of the art.

6. Figure 1 is unclear, what is the gun (targets?)? Is there any plasma? Any magnetrons? Any gases for plasma? Please improve seriously Fig. 1 or replace it with some other Figure representing the real magnetron sputtering.

7. The experimental chapter must be seriously improved regarding the magnetron sputtering parameters and also the apparatus description and error.

8. The quality of all figures must be improved to can understand the words and numbers and to better see the spectra and all drawings.

9. Please specify in text how did you identified the XRD diffraction peaks, inserting the proper references or other identifiers. Please identify all these XRD peaks properly in text.

10. Page 3 of the PDF file: “However, from Figure 2, the peaks are hard to help us to tell the difference between FeGa or Ni from the patterns, because their peaks are in the similar positions” – please insert in text how did you conclude this, based on literature or it is an authors’ assumption not supported by anything. Please do not discuss any result without proper reference or investigation.

11. Page 4 of the PDF file: “The result could be that the sputtering direction of the FeGa and Ni targets are opposite”- this is an unclear sentence. Please clarify or erase it.

12. The authors must understand that the EDX is a qualitative estimation of the composition of a film, they cannot conclude and discuss only based on the EDX indicated variation of the composition. EDX must be used maximum for semi-quantitative measurements and Fig. 3 (b) is not accurate and must be eliminated. You can introduced the EDX values in a Table with estimated uncertainties. All related discussion is not supported by trustable data.

13. The discussion on the magnetic anisotropy with increasing angle is not entirely supported by Fig. 4. Please improve the discussion.

14. The same for the discussion related to Fig. 5. The most assumptions are not supported by experimental data or by other authors works. You must improve the discussion.

15. Please improve the Conclusions according to the above observations.

Author Response

 Thanks very much for your comments on this paper. Please the response in the attachment.

Round 2

Reviewer 1 Report

Please re-read the article and correct minor typos. 

Author Response

Dear Reviewer,

Thank you again for your comments.

Zhen Wang

Reviewer 3 Report

Dear Authors,

Regarding your answer

Main comments:

1. Agree

2. Agree

3. Agree

4. Disagree. The purpose of the work is desirable to indicate in the text of the article

5. Agree

Additional comments:

1. Disagree. See lines 18, 51, etc.

2. Disagree. If you introduce a new term “mechanically magnetic”, it is necessary to explain its meaning in the text.

Author Response

(The authors gave the same response as above.)

Reviewer 4 Report

Dear Authors,

You made the suggested revisions and explained the problems in discussion. I recommend your paper to be published after another check for English translation: Vertical bar of Fig. 3: "Crystallinte" must be changed to "Crystallite"; Page 4 of PDF file: "Figuer" must be changed to "Figure". 

Author Response

Dear Reviewer,

Thank you again for your comments. We have revised our manuscript.

Zhen Wang
